# SARS-CoV-2 and Skin: New Insights and Perspectives

**DOI:** 10.3390/biom12091212

**Published:** 2022-08-31

**Authors:** Gerardo Cazzato, Eliano Cascardi, Anna Colagrande, Caterina Foti, Alessandra Stellacci, Maricla Marrone, Giuseppe Ingravallo, Francesca Arezzo, Vera Loizzi, Antonio Giovanni Solimando, Paola Parente, Eugenio Maiorano, Gennaro Cormio, Angelo Vacca, Leonardo Resta

**Affiliations:** 1Section of Molecular Pathology, Department of Emergency and Organ Transplantation (DETO), University of Bari “Aldo Moro”, 70124 Bari, Italy; 2Department of Medical Sciences, University of Turin, 10124 Turin, Italy; 3Pathology Unit, FPO-IRCCS Candiolo Cancer Institute, Str. Provinciale 142 lm 3.95, 10060 Candiolo, Italy; 4Section of Dermatology, Department of Biomedical Sciences and Oncology (DIMO), University of Bari “Aldo Moro”, 70124 Bari, Italy; 5Section of Legal Medicine, Department of Interdisciplinary Medicine, Bari Policlinico Hospital, University of Bari, 70124 Bari, Italy; 6Section of Gynecology and Obstetrics, Department of Biomedical Sciences and Human Oncology (DIMO), University of Bari “Aldo Moro”, 70124 Bari, Italy; 7Section of Internal Medicine, Department of Biomedical Sciences and Human Oncology (DIMO), University of Bari “Aldo Moro”, 70124 Bari, Italy; 8Pathology Unit, Fondazione IRCCS Casa Sollievo della Sofferenza, 71100 San Giovanni Rotondo, Italy; 9Oncology Unit IRCCS Istituto Tumori Giovanni Paolo II and Department of Interdisciplinary Medicine (DIM), University of Bari, 70124 Bari, Italy

**Keywords:** SARS-CoV-2, skin, dermatopathology, manifestation, ACE2, TMPRSS2, pandemic

## Abstract

The SARS-CoV-2 pandemic has disrupted global health systems and brought the entire globe to its knees. Although born as a disease of the respiratory system, COVID-19 can affect different parts of the body, including the skin. Reports of ongoing skin manifestations of COVID-19 have gradually multiplied, pushing researchers to investigate the etiopathogenic mechanisms underlying these phenomena in more depth. In an attempt to investigate the possible association between SARS-CoV-2, ACE2, TMPRSS2 and skin manifestations, we performed immunohistochemical investigations of the ACE2 receptor and TMPRSS2 in nine skin samples from SARS-CoV-2-positive patients compared to a cohort of healthy controls. Furthermore, after consulting public databases regarding ACE2 mRNA expression in various cell populations resident in the skin, we conducted a literature review aimed at outlining the current state of this topic. We did not find statistically different immuno-expression of ACE2 and TMPRSS2 between the group of SARS-CoV-2-positive patients (nine skin biopsies) and the control group. Regarding ACE2, major immunolabeling was present in the epidermal keratinocytes and, rarely, in the fibroblasts and in the adenomeres of the eccrine sweat glands. Regarding the immune expression of TMPRSS2, we found no significant differences between the two groups, with a weak immune staining only in some skin cytotypes. From the review of the literature, we isolated 35 relevant articles according to the inclusion criteria adopted. ACE2 appears to be a target of SARS-CoV-2, although, other receptor molecules may potentially be implicated, such as TMPRSS2. Future studies with large cases and different molecular investigative methods are needed to further elucidate the mechanisms underlying the skin manifestations of SARS-CoV-2.

## 1. Introduction

At the end of December 2019, the Chinese authorities began to report the first cases of anomalous pneumonia not related to known etiological agents, but which appeared to be very contagious and rapidly transmitted [1,2]. The epidemic soon spread everywhere, and on 11 March 2020, the World Health Organization (WHO) proclaimed the Coronavirus Disease-19 (COVID-19) as a pandemic [3,4]. By 1 August 2022, 7:50 p.m., there were 572,239,451 confirmed cases of COVID-19, including 6,390,401 deaths worldwide, and a total of 12.248.795.623 vaccine doses had been administered [5]. In Italy, from 3 January 2020 to 7:50 p.m., 1 August 2022, there were 21,059,545 confirmed cases of COVID-19 with 172,207 deaths, and a total of 51,387,413 vaccine doses had been administered [6]. Although it was quite clear from the start that COVID-19 mainly affected organs, such as the lungs and upper and lower airways [7], in the following months, extrapulmonary manifestations were increasingly reported, such as the involvement of the intestine, kidney, liver, heart, brain, skin and placenta [8,9,10,11,12,13]. Regarding skin involvement, reports of different eruption patterns became very frequent and numerous, both in the adult and pediatric population, first in China and then gradually throughout the globe [14,15,16]. Among the different cutaneous manifestations of COVID-19, erythematous rashes [17,18,19,20,21,22,23], pseudo-chilblains [24,25,26], acro-ischemia and livedoid lesions [27,28,29,30,31,32] or other cutaneous manifestations resembling pityriasis rosea, erythema elevatum diutinum, erythema multiforme and Grover disease have been reported [33,34,35,36,37,38,39,40].

ACE2, a transmembrane protein enzyme, is well known to be directly involved in the functional mechanisms of the renin–angiotensin–aldosterone system (RAAS) [41]. This physiological axis has a pivotal role in the regulation of blood pressure and in the regulation of electrolyte homeostasis. Starting with angiotensinogen produced in the liver, angiotensin I is obtained, which is converted by ACE (present in the endothelium, lung and kidney) to angiotensin II, a peptide hormone, which is then cleaved into angiotensin 1–7, which will cause vasodilation and anti-inflammatory effects [42,43]. SARS-CoV-2 virions bind to the receptor-binding domain (RBD) of S1 at ACE2, while the S2 domain mediates the fusion of viral and host cell membranes [44]. These molecular mechanisms are illustrated in Figure 1.

Thereafter, the presence of transmembrane serine protease 2 (TMPRSS2) is required, which cleaves protein S at sites S1/S2 and S’2 [45,46,47]. After that, the transcription processes orchestrated inside the infected cell allow numerous virions to be obtained, triggering an exponential growth and spread of the disease [47]. Although many studies have been conducted into the expression of ACE2 in various parts of the body [48], less evidence has been obtained at the skin level, although, in recent times, research has also been proceeding in this direction.

To investigate the possibility of penetration of SARS-CoV-2 into the skin, we conducted a study utilizing immunostaining with anti-ACE2 antibody of nine skin biopsies of patients affected by COVID-19, with demonstrated positivity to SARS-CoV-2 in RT-PCR and immunohistochemistry, and in the same number of negative control cases.

Furthermore, in an attempt to study the expression of TMPRSS2, a facilitator involved in the mechanism of ACE2 binding, we performed immunostaining with antibody against TMPRSS2.

Finally, we conducted a review of the current literature in order to trace any pathways relating ACE2 to the manifestations of SARS-CoV-2 and to attempt to develop future perspectives based on everything that this pandemic has taught us.

## 2. Materials and Methods

### 2.1. Procedure

We pooled the skin biopsy samples received at our pathological anatomy laboratory between 16 May 2020 and 10 May 2021. All patients had undergone molecular swab tests for SARS-CoV-2 on oropharyngeal swabs, using GeneXpert Dx Xpress SARS-CoV-2 RT-PCR (Cepheid, Caribbean Drive Sunnyvale, CA 94089 USA) [49], before being admitted to the Complex Operative Unit of Dermatology and Venereology of our hospital. The analytical sensitivity and specificity of these tests are reported by the manufacturers as being 100% (87/87 samples) and 100% (30/30 samples), respectively, with a detection limit of 250 copies/mL or 0.0100 plaque-forming units per milliliter [50].

All clinical data of the biopsied patients, including the severity of COVID-19, are summarized in Table 1. Symptoms that were considered mild include fever, cough, altered taste, malaise, headache and myalgia accompanied by neither dyspnea (breathing difficulties) nor radiologically detectable changes.

Symptoms that were considered moderate include oxygen saturation (SaO2) equal to or greater than 94% and/or clinical or radiological evidence of pneumonia. Symptoms that were considered severe include SaO2 < 94% or signs/symptoms of respiratory failure (IR).

Controls were chosen from a series of biopsies generally conducted during enlargements of previous non-melanoma skin cancer (NMSC) which required the removal of additional tissue that resulted in negative results. All biopsies were excisional biopsies, fixed in buffered formaldehyde at 10%, sampled according to guidelines, processed, paraffin-embedded, microtome-cut (5-micron thickness) and subjected to routine Hematoxylin/Eosin staining (H&E). They were observed with an Olympus BX-51 Optical Microscope (Shinjuku Monolith, 2-3-1 Nishi-Shinjuku, Shinjuku, Tokyo, Japan) equipped with the Olympus DP80 image-acquisition system. On all samples, IHC investigations were performed using the rabbit anti-SARS-CoV-2 spike S1 glycoprotein monoclonal antibody (MA5-36247), ThermoFisher, (Waltham, Massachusetts, USA), at pH 6 with a dilution of 1:800 and antigenic unmasking heat-induced citrate buffer epitope retrieval for enzymatic IHC analysis (as indicated by the manufacturer). Lung sections of COVID-19 subjects were used as positive controls and lung sections of swab-negative subjects as negative controls. In addition, we conducted immunostaining with recombinant anti-ACE2 rabbit monoclonal antibody [EPR4435] (ab108252) at a 1:500 dilution. We used kidney-tissue controls according to the manufacturer’s instructions. Furthermore, we performed immunostaining using recombinant anti-TMPRSS2 antibody, rabbit monoclonal, [EPR3861] (ab92323) at a 1:1000 dilution. We used a small-bowel biopsy sample as per the manufacturer’s directions.

Furthermore, all skin biopsies (18) were analyzed using GeneXpert Dx Xpress SARS-CoV-2 RT-PCR (Cepheid).

### 2.2. Immunohistochemistry Score

The expression of ACE2 and TMPRSS2 was assessed by highlighting the chromogen signal on the cytoplasm and/or cell membrane. A score was assigned, summing the different degrees of staining intensity (grade 0 = no staining; grade 1 = weak staining; grade 2 = moderate staining; grade 3 = intense staining), and added to the score for the percentage extension of the signal (score 0: <1%; score 1: 1–25%; score 2: 26–50%; score 3: 51–74%; score 4: ≥75%). The final score (sum of the two previous scores) was considered high if >3 and low if ≤3. Evaluations of immunohistochemistry sections were conducted using the Visikol platform, with the Aperio XT2 Slide Scanner and Brightfield Imaging. Acquisitions were performed at 20× and 40× magnification, and cell counts and colocalization analyses of the chromogenic signal were performed [51].

### 2.3. Statistical Analysis

For each sample, the mean and standard deviation values for the 10 fields were recorded. Normal distributions were assessed with the Kolmogorov–Smirnov test. A non-parametric Mann–Whitney test was performed for non-normally distributed values. Comparison of the means was conducted in the single groups and between the two study groups. A value of *p* ≤ 0.05 was set as statistically significant. All statistical analyses were conducted using the Prism 9.3.0 program, GraphPad software, La Jolla, CA, USA.

### 2.4. Review of Literature

A review of the literature was conducted following PRISMA guidelines using PubMed and Web of Sciences (WoS) with the words: “Coronavirus” and “COVID” in combination with each of the following: “dermatology”, “skin” and “rash” with “ACE2” and “RAAS”. Only articles published in English were selected. The last search was conducted on 1 August 2022. Eligible articles were assessed according to the Oxford Centre for Evidence-Based Medicine 2011 guidelines [52]. Review articles with or without meta-analyses, observational studies, case reports, letters to the editor and commentaries were included. Other potentially relevant articles were identified by manually checking the references of the included literature. An independent extraction of articles was performed by two investigators (G.C. and A.C.) according to the inclusion criteria. Disagreements were resolved by discussion between the two review authors. The review was conducted focusing exclusively on the relationship between ACE2/TMPRSS2 expression levels in the skin, and the presence of skin manifestations in SARS-CoV-2 patients whose positive status was ascertained by means of swab and/or molecular methods.

## 3. Results

All skin biopsy specimens were immunostained with anti-SARS-CoV-2 S1 glycoprotein monoclonal antibody and were 9/9 positive in positive patients and negative in controls (9/9). The IHC reactions carried out showed a strong positivity at the cytoplasmic level of the cells constituting the excretory portion of the eccrine sweat glands (Figure 2A,B) and, more rarely, positivity was observed in the endothelium of the small blood vessels.

In our nine cases of SARS-CoV-2 and in the controls, we found a fair degree of overlapping positivity for ACE2. A high expression (>3 in absolute value) was found in 9/9 cases (100.00%) in the samples taken from virus-positive patients, and in 8/9 subjects (88.8%) in the control group (*p* = 0.9758) (Figure 3). The immuno-expression for ACE2 was greater at the level of the epidermis (epidermal keratinocytes, brown signal), and a modest signal was also present at the cell level, constituting the adenomeres of the eccrine sweat glands (Figure 4A,B). We analyzed the ACE2 mRNA expression and ACE2-positive cellular composition in skin tissues based on public databases (GEPIA2 [53], ARCHS4 [54] and the human protein atlas [55]), which showed that ACE2 was also expressed in skin tissues as well as in the testis, kidney, colon, lung and other sites. ACE2 expression was significantly higher in keratinocytes than in other cell types in skin tissues, such as fibroblasts, endothelial cells, hair follicles, immune cells and melanocytes (Figure 5) [55].

Regarding the immunostaining for TMPRSS2, in both groups, the majority of keratinocytes were negative, with mild positive staining in eccrine sweat gland adenomeres and other cellular components, such as fibroblasts (Figure 6A,B).

In the literature review, 77 records were initially identified in the literature search, 11 of which were duplicates. After screening for eligibility and inclusion criteria, 33 publications were included (Figure 7). The majority of publications were reviews (*n* = 18), followed by original articles (*n* = 13) and case series (*n* = 4). All studies included were rated as level 4 or 5 evidence for clinical research as detailed in the Oxford Center for Evidence-Based Medicine 2011 guidelines [52]. Inclusion criteria for the review comprised the following: articles written in English; positivity of patients to SARS-CoV-2 ascertained by molecular swab; the presence of rashes in the course of COVID-19 of any clinical entity; and the use of skin biopsies.

## 4. Discussion

The SARS-CoV-2 pandemic rapidly spread around the entire globe and continues to pose a major global health challenge [1,2,3,4,5,6,7,8,9,10,11,12,13,14,15,16,17,18,19,20,21,22,23,24,25,26,27,28,29,30,31,32,33,34,35,36,37,38,39,40,41,42,43,44,45,46,47,48]. Although the involvement of the lungs and airways was immediately clear [56], as time went on, the possibility of the involvement of different organs, including the skin, became increasingly evident. Study of the molecular mechanisms by which SARS-CoV-2 enters host cells has shed light on different receptor proteins, including ACE2 and TMPRSS2 [57]. Therefore, during the pandemic months, we focused on the expression of these receptors in the most disparate body areas, including the skin, considering that skin manifestations were being increasingly reported in SARS-CoV-2-affected, or previously positive, patients [58,59,60,61,62,63,64,65,66,67,68,69,70,71,72,73,74,75,76,77,78,79,80,81,82,83,84,85,86,87,88]. In a January 2021 paper, Xiaotong X. et al. [47] conducted an elegant analysis using different molecular methods to study the expression of ACE2 in skin biopsy samples. The authors studied the expression characteristics of ACE2, starting with 18 skin samples, by means of RNA-sequencing analysis (scRNA-seq) and an immuno-expression profile using anti-ACE2 monoclonal antibody. A greater expression of ACE2 was observed at the level of epidermal keratinocytes (more basal) and at the level of eccrine glands, suggesting that this receptor in the skin could be an attachment point for the virus. Additionally, in our cases, we detected a greater immuno-expression for ACE2 in the epidermis, as well as in the adenomers constituting the eccrine sweat glands. These same areas were those stained with the anti-S1 spike glycoprotein of SARS-CoV-2 antibody. In another paper by Salamanna F. et al., the presence of ACE2 in the various body districts is summarized in detail, paying particular attention to less studied areas [48]. In the skin, for example, the authors underline that the great dilemma is to understand whether these clinical manifestations are related to an “effective” presence of virions at the cutaneous-district level, or to an immune reaction, which, among other sequelae, also determines different dermatotic patterns. There is no univocal agreement in the literature on this question, due to the different methods used (immunohistochemistry versus PCR versus RNA-seq), their limitations and the difficulty of reproducing the various results obtained by different research groups.

Other authors have also examined the molecular mechanisms underlying these manifestations [74,75,76,77,79,81,83,84,85,86,87]. For example, Sun T. et al. [78] conducted a careful analysis of 3128 patients with a laboratory diagnosis of COVID-19: skin rashes were present in 52 patients (1.66%) and single-cell RNA sequencing analysis demonstrated the colocalization of ACE2 and TMPRSS2 in the keratinocytes of the granular layer. The paper by Garduno-Soto et al. demonstrates that we are still far from being able to draw definitive conclusions [80]. Analyzing the gene expression profiles from various databases, the authors found a high presence only in the gastrointestinal tract and kidney, but not in the skin, except in the human immortalized keratinocyte HaCaT cell line. This cell line expressed detectable levels of ACE2, whereas no cell line originating in the skin expressed TMPRSS2. Another important contribution is provided by Magro C. et al. [82], who investigated the molecular mechanisms responsible for ongoing COVID-19 damage. They outline a picture whereby, starting from a microangiopathy of the pulmonary capillaries associated with a high viral load, endothelial cells die, releasing pseudovirions into the circulatory system. According to the authors, these pseudovirions dock onto the most prevalent ACE2+ endothelial cells in the skin/subcutaneous fat and in the brain, which activates a complement pathway/coagulation cascade, causing a systemic procoagulant state and the overexpression of cytokines producing a cytokine storm.

Finally, in a paper published in June 2021, Birlutiu V. et al. [88] studied 39 patients with SARS-CoV-2 who presented typical symptoms such as anosmia, ageusia, weakness and fatigue; in two cases, a skin biopsy was performed, which showed similar characteristics to other lesions described in the literature. The authors postulated a basic mechanism, mediated by ACE2, which could be common to cutaneous manifestations, anosmia, ageusia and enteritis developing during COVID-19.

Since the study of ACE2 is closely linked to that of TMPRSS2, it was important to complete the paper by also studying this other transmembrane protein receptor. Our results contribute to a still-unresolved scientific debate. In fact, our data regarding the immuno-expression of TMRPSS2 do not seem to be in agreement with authors such as Baughn et al. [89], who instead reported an increased pattern of TMRPSS2 mRNA expression in the skin of patients analyzed; the data were not confirmed by our work. On the other hand, our data are in agreement with the paper by Garduno-Soto et al. [80], which describes a near-absence of TMRPSS2 expression at the skin level, postulating that cutaneous manifestations from SARS-CoV-2 were more attributable to paracrine or endocrine (circulating) factors than to the “effective” binding of the virion to a specific skin cell type. With regard to this aspect, there are also different interpretations in the various works present in the literature [7,8,9,10,11,12,13,14,15,16,17,18,19,20,21,22,23,24,25,26,27,28,29,30,31,32,33,34,35,36,37,38,39,40,41,42,43,44,45,46,47,48,49,50,51,52,53,54,55,56,57,58,59,60,61,62,63,64,65,66,67,68,69,70,71,72,73,74,75,76,77,78,79,80,81,82,83,84,85,86,87,88], which offer different opinions, even if our work would seem to confirm that the virus is localized specifically at the level of skin cells, mainly consisting of the constituent cells in the excretory portion of the eccrine sweat glands, and to provide a rather clear molecular basis for the pathogenesis of the ongoing COVID-19 pandemic.

A paper by Liu J. et al. has already addressed this question and testifies to the possibility that ACE2 and TMRPS22 can act, at the level of the eccrine sweat glands (luminal side), as virion entry gates (highlighted by the anti S1-glycoprotein staining) so as to provide a possible proof of a direct skin trophism by SARS-CoV-2 [90].

Finally, it is important to remember that other protein receptor molecules and different factors have also been hypothesized as playing a role in causing cutaneous manifestations during infection. These include the deposition of complement cascade factors, mainly represented by C5b-9 and C4d, which have been found to be significantly expressed in the skin of patients with purpura retiform [63]. In addition, other SARS-CoV-2 non-structural proteins, such as NSP3, which have the function of blocking the host immune response and promoting a cytokine storm, have also been studied [91].

More recently, a few papers have been published in English literature about this issue; for example, Lin E. et al. [92] studied the expression of ACE2 in three different groups of conditions by immunohistochemistry: (1) patients with a dermatotic condition such as atopic dermatitis (AD), (2) patients with psoriasis and healthy patients. Furthermore, in their work, they conducted investigations of immunofluorescence, flow cytometry and quantitative RT-PCR to understand if and how ACE2 could be over-expressed at the level of epidermal keratinocytes, especially in conditions of inflammatory skin diseases. In doing so, the authors found that ACE2 expression in the skin increased via a biochemical signaling pathway mediated by the ERK protein, closely dependent on one of the major players in many skin dermatoses, Interleukin-33 (IL33). This paper has the merit of representing a further point of view that correlates the expression of the ACE2 receptor to possible mechanisms of both etiopathogenesis and SARS-CoV-2 infection.

Finally, Ganier et al. [93] conducted a study on the spatial distribution of mRNA using multiplex RNA in situ, by consulting publicly available single-cell RNA-sequencing data sets. In this work, the authors demonstrate an immuno-expression of ACE2 both at the level of epidermal keratinocytes and at the level of the vascular endothelium of the dermal plexus, suggesting that this mechanism may be the basis (still to be investigated further) for SARS-CoV-2 skin manifestations.

### Limitations

It is important to remember that the entry of SARS-CoV-2 into the skin (although confirmed by some previous studies, including our study) is only one of the possible mechanisms to explain and analyze the ongoing skin manifestations of COVID-19. Therefore, our results, which are related to the immuno-expression of ACE2 and TMPRS22, are to be considered interpretable in the light of the virion ‘direct penetration’ theory. However, in the case of the immune-mediated theory used as the basis for the explanation of SARS-CoV-2 rashes, these results would not be interpretable in this way.

## 5. Conclusions

The SARS-CoV-2 pandemic continues its march, although vaccination campaigns are managing to contain and limit its harmful effects, in terms of both morbidity and mortality. Among the different manifestations possibly related to SARS-CoV-2, cutaneous manifestations play an important role, which is not yet well understood.

The study and analysis of receptor molecules such as ACE2 and TMPRSS2 are of great importance in the attempt to elucidate the mechanisms underlying these clinical manifestations. In particular, our data seem to confirm that SARS-CoV-2 is able to “directly” infect the cells constituting the excretory portion of the eccrine sweat glands, providing a rather robust basis for the possibility of skin infection over the course of COVID-19. On the other hand, not all authors agree on this aspect, and the present results are still quite contrasting. Furthermore, whilst ACE2 and TMPRSS2 are potentially important in determining infection and therefore the dermatological clinical signs and symptoms of COVID-19, our work, together with that of others, does not seem to confirm this hypothesis. Therefore, as things stand, it would seem that SARS-CoV-2 is able to “directly” infect the skin without using these two classic “access doors” present in various other body districts, but probably able to use other protein receptors that need to be studied. Future studies, perhaps including larger case studies, are required to arrive at a definitive understanding of this strange but important aspect of the ongoing pandemic.

## Figures and Tables

**Figure 1 biomolecules-12-01212-f001:**
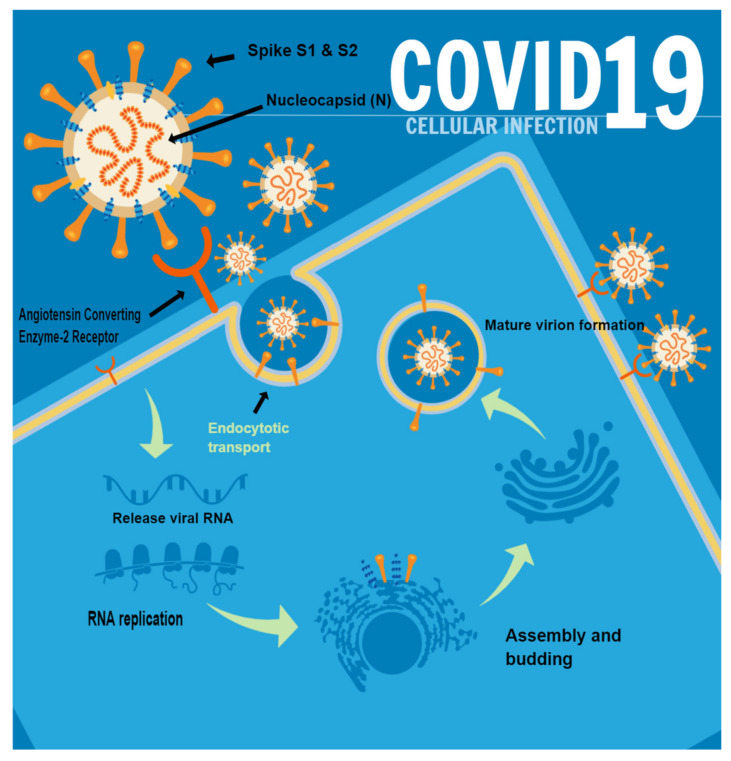
Summary of the SARS-CoV-2 spike protein-binding mechanism to the ACE2 receptor, resulting in virion entry, replication, assembly of new virions and dissemination of new viral particles. It should be noted that this mechanism has also been proposed at the level of keratinocytes and cells constituting the cutaneous eccrine sweat glands.

**Figure 2 biomolecules-12-01212-f002:**
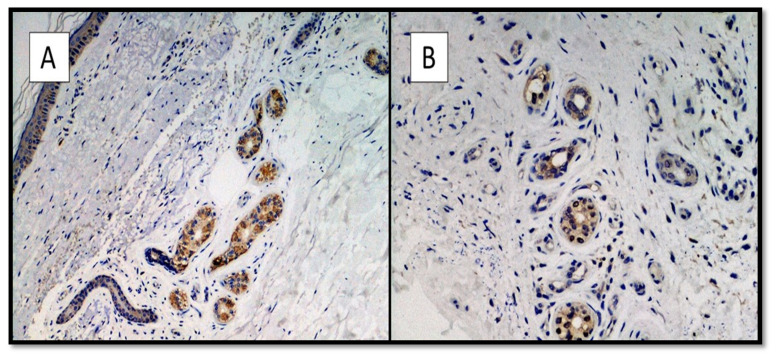
(**A**,**B**). Histological preparation for immunostaining with anti-SARS-CoV-2 S1 spike protein monoclonal antibody in a patient of the positive group. Note the granular, cytoplasmic positivity at the level of the cells constituting the eccrine sweat glands (IHC, Original Magnification: 10× and 40×).

**Figure 3 biomolecules-12-01212-f003:**
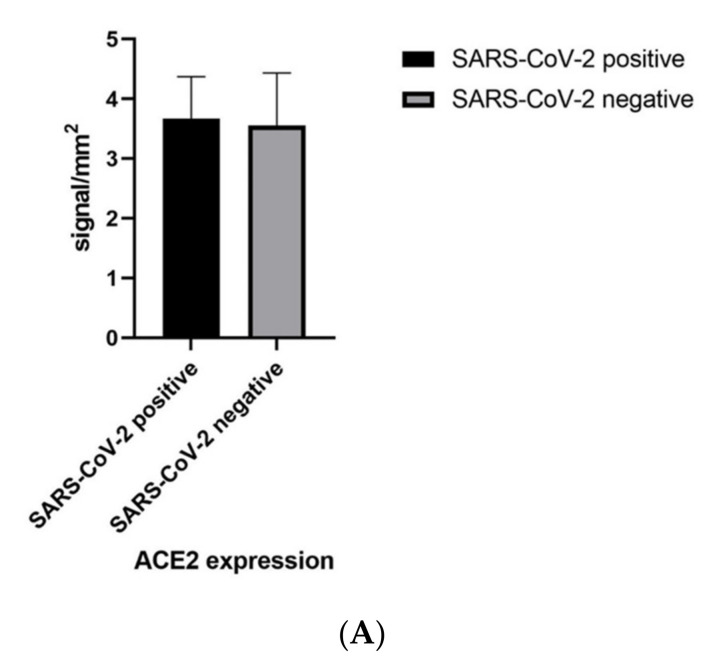
(**A**,**B**) Graphical representation of the data obtained from image analysis using Visikol related to the immunoexpression of ACE2 and TMPRSS2. Note the statistical non-significance between the two study groups (*p* > 0.05).

**Figure 4 biomolecules-12-01212-f004:**
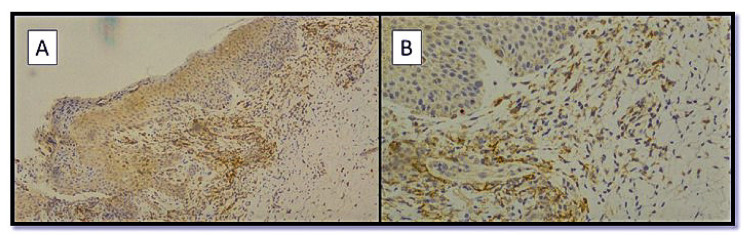
(**A**) Preparation for immunostaining with anti-ACE2 antibody from skin samples of SARS-CoV-2-positive patients. Note the focal positivity of the immunolabeling of the basal keratinocytes and of some keratinocytes of the Malpighian (spinose) and granular layer (Immunohistochemistry anti-ACE2, original magnification: 4×). (**B**) Detail of image A showing mild positivity of the ACE2 immunolabeling of some adenomeres of eccrine sweat glands and fibroblasts (Immunohistochemistry anti-ACE2, original magnification: 20×).

**Figure 5 biomolecules-12-01212-f005:**
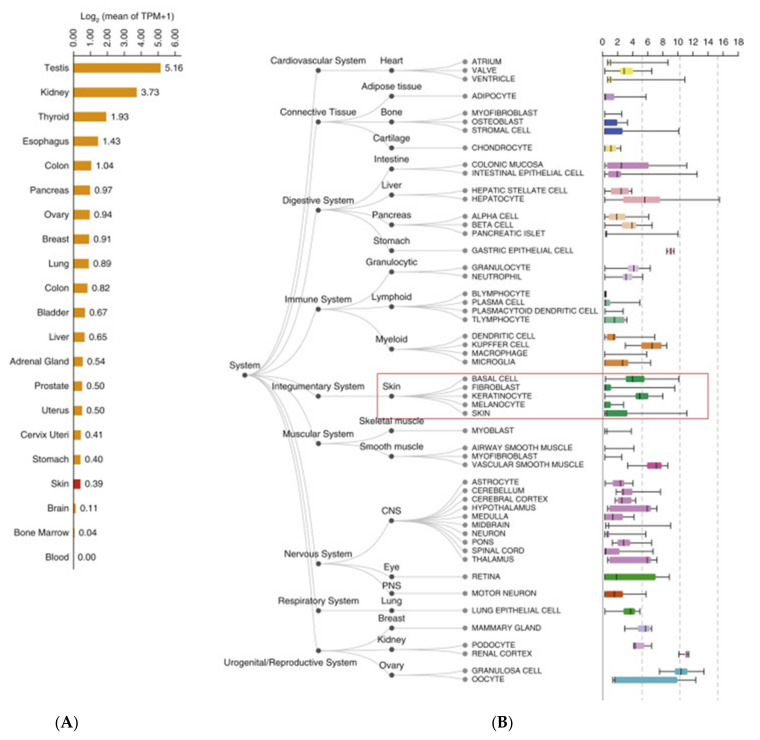
Details of ACE2 expression in skin tissues in the public databases consulted in the study. (**A**) Bar plot of ACE2 expression in normal human tissues from the GEPIA2 database. Expression level indicated by log2 (mean of TPM + 1). (**B**) Boxplot of ACE2 in different tissue types from the ARCHS4 database. TPM, transcript per million.

**Figure 6 biomolecules-12-01212-f006:**
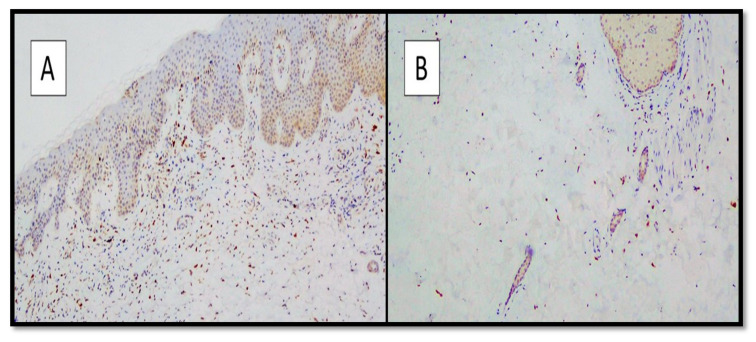
(**A**) Preparation for immunostaining with anti-TMPRSS2 antibody from skin samples of SARS-CoV-2-positive patients. Note the negativity of the immunolabeling of the epidermal keratinocytes (Immunohistochemistry anti-TMPRSS2, original magnification: 4×). (**B**) Detail of image A showing negativity of the TMPRSS2 immunolabeling of some adenomeres of eccrine sweat glands. (Immunohistochemistry anti-TMPRSS2, original magnification: 20×).

**Figure 7 biomolecules-12-01212-f007:**
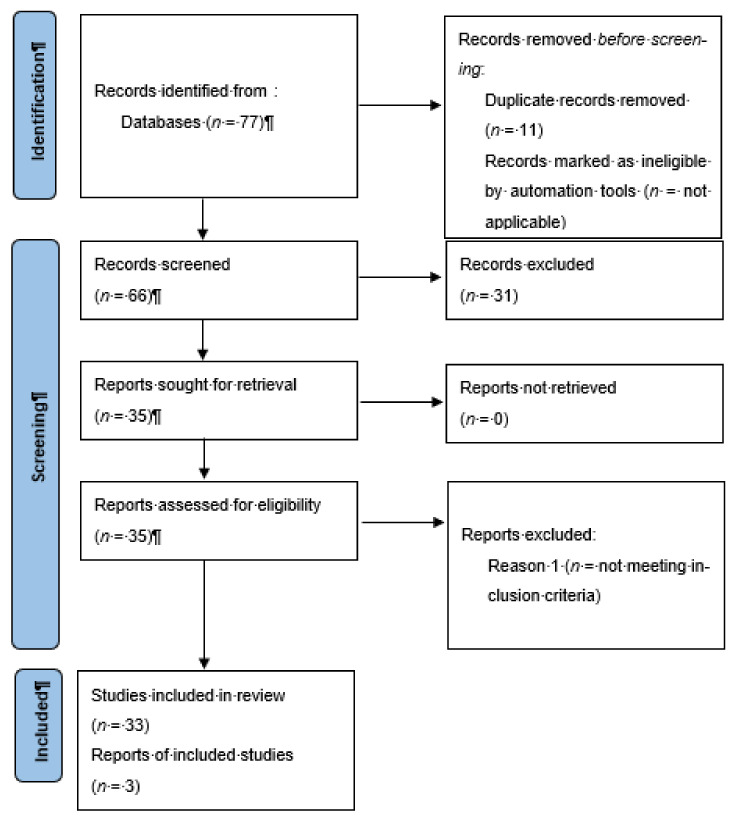
Literature search and article selection following PRISMA guidelines.

**Table 1 biomolecules-12-01212-t001:** Clinical characteristics of the patients enrolled in this study.

Number of Patient	Age	Gender	Type of Dermatological Manifestation	Severity of COVID-19
1	62	M	Erithema Pernio	Mild symptoms
2	14	M	Erithema Pernio	Mild/asymptomatic
3	11	M	Erithema Pernio	Mild/asymptomatic
4	41	M	Folliculitis-like rash	Moderate symptoms
5	17	F	Erithema Pernio	Mild symptoms
6	29	M	Urticarial rash	Mild symptoms
7	52	F	Varicelliform-like rash	Severe symptoms
8	68	F	Urticarial rash	Moderate symptoms
9	16	M	Erithema Pernio	Asymptomatic

## Data Availability

Not applicable.

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
