# Peer review of "SARS-CoV-2 and Skin: New Insights and Perspectives"

_biomolecules, 2022, doi:10.3390/biom12091212_

Round 1

Reviewer 1 Report

The paper is well prepared and can be accepted after minor revision:

the novelty of the work it is to be clearly stated.

The introduction is to be extended.

How can you be certain that the skin problems with the patients are directly related to SARS-CoV-2?

The statistical analysis is to be detailed.

The molecular mechanisms are to be described with more details.

The paper is to be checked against misprints and grammatical mistakes.

Author Response

Reviewer n'1: The paper is well prepared and can be accepted after minor revision: the novelty of the work it is to be clearly stated. The introduction is to be extended. How can you be certain that the skin problems with the patients are directly related to SARS-CoV-2?

Answer n'1: Dear Reviewer n'1, first of all, thank you very much for your kind congratulations to our manuscript. We improved introduction of our paper in order to add some informations about the potential ethiological mechanisms of cutaneous manifestations of SARS-CoV-2: direct penetration vs immunomediated mechanisms. Futhermore, we improved the lengh of introduction and discuss this important aspect. Thanks again.

Reviewer n'1:

The statistical analysis is to be detailed.

The molecular mechanisms are to be described with more details.

The paper is to be checked against misprints and grammatical mistakes.

Answer n'2: Dear reviewer n'1, thanks again. We add some details about statistical analysis, molecular mechanisms and, finally, we corrected paper against typos.

Thanks again

the authors

Reviewer 2 Report

The study is well performed and interesting however it suffers from two drawbacks. First the number of patients and controls is too small to fully support the  conclusion/ suggestion that SARS-Cov2 may infect cells through mechanisms different from binding to ACE2. Second the manuscript is too wordy which makes difficult its reading and understanding. My advice is therefore to be more prudent in the conclusion and to clean the manuscript from redundant sentences and useless citations.

Author Response

Dear Reviewer n'2, first of all, thank you very much for your wonderful compliments which we are very proud of. We perfectly understood his point of view and, therefore, we decided to take two corrective actions: 1. we have added to the paragraph "limitations" the discourse relating to the small sample size of our study which, therefore, does not allow generalizations but only a starting point for future and extensive studies. Furthermore, in agreement with all the authors, we have decided to eliminate the "conclusions" as they are already redundant with what has been stated above. Finally we have eliminated some parts of the paper to make it more fluent and understandable. We hope it will go well. A warm greeting

Round 2

Reviewer 2 Report

The paper has improved but the significance of its content remains low.

Author Response

Thanks for reviewing our manuscript.